# Efficacy of Poloxamer 188 in Experimental Myelosuppression Model Induced by Carboplatin in CBA Mice

**DOI:** 10.3390/ijms26157081

**Published:** 2025-07-23

**Authors:** Arina V. Kholina, Natalya A. Borozdina, Victor A. Palikov, Evgeniy S. Mikhaylov, Irina N. Kravchenko, Renata A. Dalevich, Irina A. Pakhomova, Ekaterina N. Kazakova, Maria A. Timchenko, Alexander Ye. Yegorov, Maxim V. Molchanov, Artem M. Ermakov, Olga Y. Antonova, Olga Y. Kochetkova, Natalia M. Pankratova, Anton N. Pankratov, Elena V. Arshintseva, Sergey Y. Pushkin, Igor A. Dyachenko, Arkadiy N. Murashev

**Affiliations:** 1Branch of the Shemyakin-Ovchinnikov Institute of Bioorganic Chemistry, Pushchino 142290, Russia; bervinova@bibch.ru (A.V.K.); borozdina@bibch.ru (N.A.B.); vpalikov@bibch.ru (V.A.P.); mikhaylov@bibch.ru (E.S.M.); ikravchenko@bibch.ru (I.N.K.); dalevich@bibch.ru (R.A.D.); pakhomova@bibch.ru (I.A.P.); katerina_scoryh86@mail.ru (E.N.K.); dyachenko@bibch.ru (I.A.D.); murashev@bibch.ru (A.N.M.); 2Institute of Theoretical and Experimental Biophysics (RAS), Pushchino 142290, Russia; a.yeg0rov@yandex.ru (A.Y.Y.); lvlaks.m@gmail.com (M.V.M.); ao_ermakovy@rambler.ru (A.M.E.); olga.antonova.iteb@gmail.com (O.Y.A.); o.y.kochetkova@gmail.com (O.Y.K.); 3Institute of Mathematical Problems of Biology (RAS)—The Branch of Keldysh Institute of Applied Mathematics of Russian Academy of Sciences, Pushchino 142290, Russia; natpan1974@mail.ru (N.M.P.); pan@impb.ru (A.N.P.); 4Medical Emulsions LLC, Pushchino 142290, Russia; micolina@mail.ru (E.V.A.); laboratorypushkin@yandex.ru (S.Y.P.)

**Keywords:** Poloxamer 188, myelosuppression, carboplatin, 2,3-bisphosphoglycerate, CBA mice

## Abstract

Poloxamer 188 is a polymer that is used as a carrier and stabilizer of pharmacological agents. It has been demonstrated to enhance red blood cell and hemoglobin levels in healthy animals and in select clinical cases. The objective of this study was to assess the efficacy of Poloxamer 188 in CBA mice when administered repeatedly in the carboplatin-induced myelosuppression model. The mice were administered carboplatin once at a dose of 100 mg/kg, and then Poloxamer 188 was orally administered daily at doses of 10 mg/kg, 100 mg/kg, 500 mg/kg, and 1000 mg/kg for 7 and 21 days. Poloxamer 188 at a dose of 1000 mg/kg was found to bring the level of 2,3-bisphosphoglycerate in red blood cells close to control level (*p* = 0.1331 for the control group compared to Poloxamer at a dose 1000 mg/kg) already from day 8 of the study and in bone marrow resulted in regulation of genes responsible for hematopoiesis. G-GSF at day 8 and TNFα at day 22 gene expression was significantly decreased by 54% (*p* = 0.012) and 16% (*p* = 0.024), respectively, with Poloxamer 188 administration at a dose of 100 mg/kg. Additionally, in the bone marrow, the treatment was seen to exert a positive regulatory effect on the genes responsible for hematopoiesis. These findings are consistent with the observed increase in red blood cell by 6.7% (*p* = 0.001), hemoglobin by 4.7% (*p* = 0.0053), and reticulocyte percentage by 53.6% (*p* < 0.0001) following Poloxamer 188 administration at a dose of 1000 mg/kg in CBA mice with myelosuppression.

## 1. Introduction

Myelosuppression (MYS) is a common and anticipated consequence of cytotoxic chemotherapy. Myelosuppression during cytostatic treatment has been demonstrated to reduce the quality of life of patients and is potentially life-threatening due to complications [1]. Some patients demonstrate a swift recovery from myelosuppression following the conclusion of chemotherapy. However, there are instances where residual myelosuppression persists for an extended period, manifesting as decreased reserves and impaired hematopoietic stem cell renewal [2].

The primary therapeutic intervention for myelosuppression is the administration of recombinant human erythropoietin [3]. In the case of younger patients, allogeneic hematopoietic cell transplantation is undertaken prior to the commencement of immunosuppressive therapy. Elderly patients and young patients without a hematopoietic cell donor receive full-dose immunosuppressive therapy with eltrombopag, horse/rabbit anti-thymocyte globulin, cyclosporine A, and prednisone. Supportive treatment includes infection prophylaxis and transfusion of leukoreduced red blood cells to maintain hemoglobin if it is less than 7 mg/dL or platelets if less than 10,000/μL or less than 50,000/μL with active blood loss [4].

Poloxamer 188 (P188) (Figure 1) is a biocompatible block copolymer comprising repeating units of polyethylene oxide and polypropylene oxide. Given its amphiphilic nature and high hydrophilic-lipophilic balance value, P188 is employed as a stabilizer and emulsifier in numerous cosmetic preparations [5], as well as a carrier in pharmacological preparations [6], including perfluorinated compound emulsions [7,8].

Erythropoietin and erythropoietin-stimulating drugs are used for patients with tumors to prevent myelosuppression. However, there are reports that using erythropoietin to treat cancer can make tumors grow faster and reduce survival rates [10,11,12]. Erythropoietin is not recommended for people with kidney failure. There have been reports of an increased risk of heart problems due to increased hemoglobin [13]. Thus, there is a need to find alternative drugs to prevent myelosuppression caused by antitumor agents. The choice of therapy is strictly individualized.

Poloxamer 188 (P188, comprising 80 wt% polyethylene oxide, with a molecular weight of 8400 g/mol) was initially approved by the FDA for the reduction of blood viscosity. Recently, its amphiphilic nature has been employed to stabilize cell membranes under diverse stress conditions, including models of stroke, myocardial infarction, sickle cell anemia, and Duchenne muscular dystrophy [14,15,16,17].

P188 has been shown to improve erythrocyte survival by increasing membrane stability. Such properties of Poloxamer 188 are being considered for long-term blood storage [18]. P188 has membrane-protective properties, is not among the classical cytoprotectors or hematopoiesis stimulators, making it a unique target for study in the context of reducing the toxicity of alkylating agents.

P188 has exhibited a favorable safety profile with minimal adverse effects in long-term human clinical trials. The Safety Data Sheet indicates that the LD50rat for Kolliphor^®^  P 188 Geismar is greater than 5000 mg/kg in acute oral toxicity [19]. Previously, an acute toxicity study of P188 was conducted on male and female rats [7]. The animals showed no signs of intoxication when P188 was administered intravenously as a single dose of 800 mg/kg and 1330 mg/kg, and no local irritant effects were observed. Other studies on the efficacy of P188 have used doses of 800 mg/kg Poloxamer for mice when administered intravenously [20], 400 mg/kg for intravenous administration in rats [21]. For mice in the ischemia model, doses of 200–800 mg/kg by intravenous administration were used [22]. Therefore, it was decided to use the maximum oral dose of 1000 mg/kg in the efficacy study. In single clinical cases, an increase in erythrocyte concentration after Poloxamer administration was shown [23].

E.V. Arshintseva and S.Y. Pushkin highlight the impact of P188 administration on the elevation of red blood cell numbers and hemoglobin concentration in the blood of healthy animals [23]. Nevertheless, preclinical investigations into the stimulatory activity of P188 have yet to be conducted to examine the dose-dependent impact of P188 administration in an anemia model. Furthermore, there is a paucity of studies that reflect the metabolic features of the development of anemia induced by myelosuppression, despite this being a highly characteristic feature of impaired blood gas transport [24]. The objective of this study was to examine the efficacy of P188 in an experimental model of myelosuppression induced by carboplatin in CBA mice.

## 2. Results

### 2.1. Hematological Analysis

In male CBA mice treated with carboplatin at a dose of 100 mg/kg once intraperitoneally, the white blood cell count increased on day 8 of the study compared to the control, and relative lymphopenia was observed (Figure 2). Red blood cell count, platelet count, hemoglobin concentration, and reticulocytes were significantly decreased after carboplatin administration at day 8. Administration of P188 at a dose of 500 mg/kg and 1000 mg/kg for 7 days increased the reticulocytes compared to the MYS group. P188 at a dose of 100 mg/kg, 500 mg/kg, and 1000 mg/kg on day 8 restored hemoglobin levels relative to the MYS group. Administration of P188 at a dose of 1000 mg/kg increased red blood cell count relative to the MYS group.

On day 22 of the study, animals in the MYS group still had a decreased red blood cell count. In all groups receiving P188 for 21 days, red blood cell count was significantly higher than in the MYS group. After 21 days of P188 administration at doses of 10 mg/kg, 500 mg/kg, and 1000 mg/kg, hemoglobin levels significantly increased compared to the MYS group. Platelet count and reticulocytes recovered independently in the model group after carboplatin administration; relative lymphopenia and granulocytosis on day 22 were not observed in all groups.

At days 8 and 22, absolute and relative neutrophilia and relative lymphopenia were observed in the MYS group (Figure 3). Animals receiving P188 at doses of 100 mg/kg, 500 mg/kg, and 1000 mg/kg for 7 days showed relative neutrophilia compared to the MYS group.

On day 22, relative neutropenia was observed in all groups receiving P188. The percentage of lymphocytes was restored on day 22 of the study when P188 was administered at all doses compared to the MYS group.

The model group showed no change in monocyte percentage. However, when P188 was administered at a dose of 1000 mg/kg for 7 days, there was a significant increase in monocytes. At doses of 100 mg/kg and 500 mg/kg, there was a significant decrease compared to the control group.

In all groups receiving carboplatin, there was an increase in the granulocytes and a decrease in the lymphoid cell percentage on day 8 compared to the control group, and there was also a trend toward a decrease in the erythroid cell percentage (Figure 4). On day 22, the carboplatin-treated groups also had a decreased erythroid cell percentage, and the granulocyte percentage remained elevated relative to the control group. There was an increase of lymphoid cells in the groups MYS, MYS + P188 100 mg/kg, and MYS + P188 1000 mg/kg. On day 22, an increase in monocytes percentage was observed in the groups administered P188 at doses of 500 mg/kg and 1000 mg/kg.

### 2.2. 2,3-BPG Erythrocyte Levels

The level of 2,3-bisphosphoglycerate (2,3-BPG) in the MYS group on day 8 was significantly lower than in the control group, and when P188 was administered at a dose of 1000 mg/kg, the level of 2,3-BPG returned and even slightly exceeded the level of 2,3-BPG in the control group (Figure 5). At day 22, changes in 2,3-BPG levels were no longer observed between groups.

### 2.3. Bone Marrow Gene Expression

On day 8, a weak increase of TPO gene expression was observed in MYS group and groups with P188 administration at doses of 10 mg/kg and 100 mg/kg (Figure 6). A 1.8-fold increase of c-Mpl gene expression was observed in the MYS + P188 10 mg/kg group, while in the other groups the increase was less pronounced compared to the control group. On day 8, for the G-GSF and TNFα genes, there was a decreased expression observed compared to the control group. On day 22 of the study, there was a trend toward decreased expression of TPO and c-Mpl, and there was a significant decrease in TNFα and TPO gene expression in the group with P188 administration at a dose of 500 mg/kg. On day 22, an increase in G-GSF expression was observed in groups with P188 administration at doses of 10 mg/kg and 100 mg/kg.

## 3. Discussion

Myelosuppression is a common side effect due to long-term treatment with cytostatic drugs, including carboplatin. Myelosuppression during chemotherapy is mostly transient, but sometimes prolonged myelosuppression contributes to the development of aplastic anemia [25]. A model of myelosuppression induced by carboplatin was chosen to study myelosuppression. Carboplatin is known for its toxicity profile including myelosuppressive effects. Administration of carboplatin to mice and other animal species also produced leukopenia and thrombopenia [26,27]. We modeled experimental myelosuppression by a single administration of carboplatin at a dose of 100 mg/kg to CBA mice.

No abnormalities at clinical examination were observed in the animals during the 22-day period. Hematological analysis of the blood on days 8 and 22 showed signs of the development of mild anemia, such as reduced red blood cell, hemoglobin, and reticulocyte counts. Similarly, Woo et al. observed a decrease in hemoglobin, red blood cells, and reticulocytes in rats 8–12 days after intravenous administration of carboplatin at a dose of 60 mg/kg [28]. Authors [29] observed granulocytopenia and thrombocytopenia in mice 7–12 days after carboplatin administration, although we observed granulocytosis in our study. However, platelet count and reticulocytes recovered in the model group after carboplatin administration on day 22 of the study. Fornari et al. also reported recovery of reticulocytes in the blood after a single administration of carboplatin to mice [30].

In the bone marrow differential, there is a decrease in the lymphoid cells and erythroid cell percentage and an increase in the granulocytes percentage. Thus, by a single intraperitoneal administration of carboplatin at a dose of 100 mg/kg to CBA mice, we achieved experimental myelosuppression manifested by bone marrow failure, thrombocytopenia, relative lymphopenia, and anemia (decrease in red blood cell count and hemoglobin) [31].

Day after induction of myelosuppression, P188 was administered daily by oral gavage at doses of 10 mg/kg, 100 mg/kg, 500 mg/kg, and 1000 mg/kg for 7 and 21 days. When P188 was administered for 7 and 21 days, there was an increase in red blood cells, hemoglobin, and reticulocyte counts and a recovery of neutrophil counts in carboplatin-treated mice. In the bone marrow, P188 administration tended to restore granulocytes and monocytes percentage in bone marrow.

We quantitatively analyzed phosphate compounds in red blood cells using NMR spectroscopy because metabolic disorders occur in the development of anemia [32,33,34]. Of greatest interest in anemia is 2,3-BPG, which binds to partially oxygenated hemoglobin, allowing red blood cells to more efficiently release the remaining oxygen molecules [35]. In the MYS group, we observed a decrease in 2,3-BPG in red blood cells on day 8. However, in the MYS group we found a decrease in lactate and 2,3-BPG level. Platinum-based drugs, including carboplatin, can cause damage to erythrocyte membranes leading to potassium and sodium ion imbalance and changes in intracellular pH [36]. This directly affects the activity of bisphosphoglycerate mutase, a key enzyme in the synthesis of 2,3-BPG [37,38].

When P188 was administered, a slight increase in lactate and 2,3-BPG concentration was observed, which may indicate the enhanced oxyhemoglobin dissociation [35]. It is also known that the concentration of 2,3-BPG decreases in old red blood cells [39]. It is likely that the decrease in 2,3-BPG in the MYS group is associated with an increased old red blood cell percentage formed and a decreased number of erythroid lineage cells in the bone marrow after carboplatin treatment. When P188 was administered at a dose of 1000 mg/kg, an increase in the number of red blood cells was observed and the number of reticulocytes was restored on day 8, as was an increase in the 2,3-BPG concentration. We believe that P188, embedding into hydrophobic regions of damaged membranes, restores their integrity, prevents ion leakage, and stabilizes intracellular pH. This mechanism probably explains the preservation of physiological level of 2,3-BPG in animals treated with Poloxamer P188 against carboplatin.

The study of gene expression in a model of carboplatin-induced myelotoxicity followed by P188 therapy provided important data that emphasize the role of this drug in modulating key biological processes. We adhere to the hypothesis that changes in gene expression reflect the compensatory mechanisms of the organism in response to chemotherapy-induced damage and recovery under the influence of P188.

One significant finding was the altered expression of genes related to hematopoiesis, such as TPO and c-MPL [40,41,42]. On day 8, their expression significantly increased, especially in animals treated with P188, indicating the activation of hematopoiesis restoration processes. However, by day 22 the expression of c-MPL and TPO decreased, which was most pronounced in groups with high doses of P188. Zhao et al. observed a 2-fold decrease in c-MPL gene expression and a more than 4-fold decrease in TPO expression at 14 and 28 days after cyclophosphamide administration in a murine model of myelosuppression [43]. This may be due to the restoration of normal blood cell counts in animals from these groups and the inclusion of negative feedback mechanisms regulating excessive proliferation.

G-GSF gene expression also demonstrates a dose-dependent effect of P188. The G-GSF gene stimulates the production of granulocytes, and promotes the mobilization of hematopoietic stem cells into the peripheral blood. Its expression is upregulated in infections and it is clinically used in the treatment of leukopenia [44]. In our study, we observed a significant inhibition of G-GSF upregulation on the background of carboplatin administration in combination with P188 on day 8 of the study, although there was a significant increase in the proportion of granulocytes in the bone marrow. We assume that the expression of G-GSF gene was higher in earlier terms of myelosuppression modeling.

The decrease in TNFα gene expression in bone marrow on both day 8 and day 22 after modeling of myelotoxicity by carboplatin can be explained by several factors. First, carboplatin, as a cytostatic drug, directly inhibits the proliferation and activity of bone marrow cells, including stromal and immune cells, which are the main sources of TNFα [45,46]. This leads to a decrease in gene expression at early stages. Secondly, by day 22, recovery processes begin in the bone marrow, which is accompanied by a decrease in inflammatory reactions initiated by myelotoxicity [46,47]. This naturally reduces the need for TNFα production. In addition, P188 likely contributes to cell membrane stabilization and reduced inflammation in the bone marrow, which further suppresses TNFα expression.

It should be noted that by taking two time points (days 8 and 22), we were not able to fully explore the dynamics of gene expression changes. We chose time points for this study based on a pilot study, where we observed maximal manifestation of myelosuppression (decrease in erythrocytes and hemoglobin primarily) 7 days after carboplatin administration. This limits our conclusions, and we can only speculate how the test object would have affected gene expression in the first days or 2 weeks after the end of the experiment. It is worth noting that the experiment duration of 22 days was chosen based on the results of the pilot study. A preliminary evaluation of carboplatin on hematologic blood parameters revealed a gradual decrease in erythrocytes by day 12 and a simultaneous sharp increase in reticulocytes. Already by day 21, the hematologic parameters were restored in animals from the model group.

Undoubtedly, there are obstacles in successfully translating the findings to humans, as mice have faster metabolic processes and higher rates of cellular recovery, which may lead to different duration and degree of myelosuppression. Thus, mice have smaller erythrocyte size, a higher proportion of circulating reticulocytes or polychromasia, a low proportion of neutrophils in peripheral blood and a high proportion of lymphocytes in peripheral blood and bone marrow, variable leukocyte morphology, physiologic splenic hematopoiesis and iron storage, more numerous and shorter-lived erythrocytes and platelets [48]. Nevertheless, we believe that the results of the experiment on mice have translational potential, as we initially observed symptoms typical of myelosuppression, such as decreased hemoglobin, red blood cell, and reticulocyte levels. The authors [49,50] agree that hematopoiesis in mice would be useful when considering the results from a translational perspective.

In the current study, we included two time points for hematologic and bone marrow data collection, 7 and 21 days after myelosuppression induction. In this case, we were able to demonstrate recovery in the model group of hemoglobin and reticulocytes in hematologic analysis, as well as recovery of 2,3-BPG levels in erythrocytes. However, 7 days after myelosuppression, we were still able to follow the effect of carboplatin and observed the manifestation of myelosuppression. Although we could not identify a clear dose-dependent pattern in the study, there was pronounced efficacy with respect to hemoglobin, erythrocyte, and reticulocyte levels at 500 mg/kg and 1000 mg/kg doses of P188. No significant differences were found in the percentage of granulocytes between groups with P188 administration at different doses. The 100 mg/kg dose was indeed more effective in reducing TNFα on day 22 and G-CSF expression levels on day 8. In terms of clinical parameters, a significant decrease in WBC count and lymphocyte percentage was observed in this group. However, no benefit was seen in hematopoietic parameters in mice treated with P188 at a dose of 100 mg/kg, nor with higher doses of 500 mg/kg or 1000 mg/kg. We consider the dose of P188 1000 mg/kg to be effective, but it is worth considering also the dose of 500 mg/kg to be effective, because with it we observed a significant recovery of reticulocytes and hemoglobin levels after one week of administration. Also, the dose of 100 mg/kg may be interesting for translational results. We realize that further NMR studies of 2,3-BPG levels in erythrocytes at P188 doses of 100 mg/kg and 500 mg/kg are required. It should also be noted that we used P188 from BASF Corporation, which has an average molecular weight of 7680–9510 g/mol.

To improve the experiment in the future, it is worth including a more detailed analysis of cellular dynamics, as well as expanding the panel of studied genes and proteins to obtain a more complete picture of the recovery processes under the action of P188. Further studies on the in vivo toxicity of P188 with repeated administration are needed. At the moment, the SDS has data only on acute oral toxicity [19]. In this case, we see the need to check all parameters: hematology, clinical biochemistry and urinalysis, weighing, food and water intake. Reticulocytes should also be determined in bone marrow smears. Also, limitations of the study are related to sex differences in the response of hematologic parameters to chemotherapy with platinum-containing drugs [51]. In our study, we only used male CBA mice in the carboplatin-induced myelosuppression model. Further efficacy studies on carboplatin-induced myelosuppression must include animals of both sexes.

## 4. Materials and Methods

### 4.1. Animals

The study used 72 male CBA mice aged 7–8 weeks old at the start of P188 administration. Specific pathogen-free (SPF) animals were obtained from the laboratory animal nursery, branch of the Shemyakin–Ovchinnikov Institute of Bioorganic Chemistry of the Russian Academy of Sciences (BIBCh RAS), Pushchino. Animals were kept for 14 days in a two-corridor barrier area of the BIBCh RAS facility, in groups in type III cages (820 cm^2^) for acclimatization. All animal procedures were reviewed and approved for regulatory compliance by the Institutional Animal Care and Use Committee (IACUC) of BIBCH RAS (Protocol No. 939/23 dated 23 June 2023).

During the experiment, animals were housed individually in type IV cages (370 cm^2^). Temperature and humidity were controlled automatically using the Eksis Visual Lab system (PRAKTIK-NTS, AO, Zelenograd, Russia). There was an automatic change of day and night period (08:00–20:00—“day”, 20:00–08:00—“night”) and at least 12 times change of air volume in the room per hour. Autoclaved bedding (LIGNOCEL BK 8/15, JRS, Rosenberg, Germany) was used in the cages. The animals received Velaz FORTI 1324 maintenance diet (Altromin Spezialfutter GmbH & Co KG, Im Seelenkamp 20, D-32791 Lage, Germany) and Milli-RO Millipore filtered tap water ad libitum for laboratory mice and rats. During the experiment, the animals were given nesting material SAFE Nesting Small (JRS, Rosenberg, Germany) or red transparent polycarbonate houses (Ugo Basile, Gemonio, Italy). Clinical examination was performed 2 times a day (morning and evening) to monitor the deterioration of the animals’ condition. Body weight was monitored daily, and in case of a 20% decrease in body weight from the initial weight, euthanasia of the animal in a CO_2_ chamber was scheduled.

Animals without clinical signs of impairment were randomized into 6 groups of 12 animals each on the basis of body weight (Table 1). Group size was set up after the sample size Power Calculation http://www.biomath.info/power/index.html, accessed on 05.08.2023) at 80% power and 5% alpha values in unpaired *t*-test use.

### 4.2. Myelosuppression Induction

Carboplatin was used to model experimental myelosuppression induced by chemotherapy by a single intraperitoneal injection in CBA mice. The modeling of myelosuppression was performed in groups 2–6 by intraperitoneal injection of carboplatin (Federal State Budgetary Institution “N.N. Blokhin National Medical Research Center of Oncology. Blokhin National Medical Research Centre of Oncology of the Ministry of Health of the Russian Federation) at a dose of 100 mg/kg, once, volume 10 mL/kg. Group 1, the control group, was injected with a carrier (purified Milli-RO water) in the volume of 10 mL/kg.

### 4.3. Poloxamer 188 Administration

On the next day after induction of experimental myelosuppression, animals in groups 3–6 were started on P188-Kolliphor^®^ P 188 Geismar (poly (ethylene glycol)-block-poly (propylene glycol)-block-poly (ethylene glycol), Poloxamer 188 x, z-80; y-27, BASF Corporation 8404 River Rd, Geismar, Ascension Parish, LA, USA) at doses of 10 mg/kg, 100 mg/kg, 500 mg/kg and 1000 mg/kg daily by probe into the stomach in a volume of 5 mL/kg. Groups 1–2 were administered the carrier—purified water Milli-RO (Millipore, Darmstadt, Germany) in a similar manner and volume. The administration was continued for 7 days in 6 animals of each group and for 21 days in the remaining 6 animals. The animals were examined weekly for clinical abnormalities in the cages, by hand and in the open field.

### 4.4. Necropsy

On days 8 and 22, animals were euthanized by anesthesia (Telazol^®^ (Zoetis, Parsippany, NJ, USA)/Xilanit^®^ (Nita-Farm, Saratov, Russia) mixture intramuscularly), followed by terminal blood sampling from the caudal vena cava for clinical pathology. Blood was collected from the inferior vena cava at laparotomy using a syringe with a 22G needle (0.7 × 40 mm^2^).

### 4.5. Hematological Analysis

Blood samples at necropsy (~0.1 mL) were collected in Microvette^®^ tubes (SARSTEDT, Nümbrecht, Germany) containing K3EDTA. Hematological analysis was performed on the day of blood collection using a Mythic 18 hematology analyzer (C2 DIAGNOSTICS S.A., Grabels, France). For reticulocyte counting, ~50 μL of blood was placed in a tube containing 1% cresyl blue dye for staining, followed by a blood smear. Manual reticulocyte counting was performed among 1000 red blood cells in the blood smear using an AxioScope. A1 light microscope (Carl Zeiss, Oberkochen, Germany). For white blood cell differential, a blood smear was taken from 100 leukocytes in the smear, followed by Papenheim staining.

### 4.6. Bone Marrow Differential

Bone marrow from all animals was collected by aspiration from the right femur immediately after euthanasia and placed in a centrifuge tube for smear preparation. A minimum of two slides were prepared per animal. Bone marrow smears were examined by light microscopy using an AxioScope. A1 microscope (Carl Zeiss, Oberkochen, Germany). The number of erythroid cells, myeloid progenitor/granulocytic cells, monocytes, and lymphoid cells was counted.

### 4.7. ^31^P NMR Spectroscopy of Washed Red Blood Cells

At necropsy in all animals of CTRL, MYS, and MYS + P188 1000 mg/kg groups, an additional 500 μL of blood was taken and placed in Microvette^®^ tubes (SARSTEDT, Nümbrecht, Germany) with K3EDTA, centrifuged (1500 g, 4 °C, 15 min) to remove plasma. Red blood cells were washed twice with washing buffer (154 mmol/L NaCl, 4.3 mmol/L KCl). Wash buffer was added to washed RBC up to 550 μL, and the red blood cell suspension was placed in an NMR tube. An inner capillary containing D_2_O as a reference lock was also inserted into the NMR tube.

One-dimensional (1D) ^31^P NMR spectra were obtained on a Bruker 600 AVANCE III NMR spectrometer (Bruker BioSpin, Rheinstetten, Germany, The Core Facilities Centre of Institute of Theoretical and Experimental Biophysics of the RAS). All measurements were performed at 298 K. The pulse sequence used in the experiments is a standard one from the Bruker pulse sequence library—ZG. The operating frequency for the ^31^P isotope nucleus was 242 MHz, and the spectrum width was 300 ppm. Acquisition was pursued for a 30 min period (880 scans) to obtain sufficient signal-to-noise ratio. Rotation was used for the sample. The spectra were processed in the TOPSPIN program (Bruker, Rheinstetten, Germany). Chemical shifts of 2,3-bisphosphoglycerate (2,3-BPG) (chemical shift: 3.3 ppm for 3-P and 2.6 ppm for 2-P) were used as internal chemical shift references.

### 4.8. Bone Marrow Gene Expression

At necropsy, bone marrow was collected from the left femur of all animals and placed in a tube containing EverFresh preservative, incubated for 30 min, and stored in a refrigerator at −20 °C. Randomization using a random number table and simple blinding were used in the analysis. Total RNA was isolated from the samples using the innuPREP DNA/RNA Mini Kit (Analytik Jena, Jena, Germany). Synthesis of complementary DNA (cDNA) was performed using the MMLV RT Kit (Evrogen, Moscow, Russia). Real-time PCR was performed using primers specific for the genes BRCA1, BRCA2, ERCC1, c-Mrl, TPO, JAK2, Slc13a5, p53, G-CSF, SDF-1, SCF, TNF-a, MIP-1a, TGF-B and the reference genes EF1a and RPL13a. Bone marrow cell cDNA from the CTRL group was used as a reference sample. The reaction was performed using 5X Taq polymerase-based PCR mix with hot start qPCRmix-HS SYBR + LowROX (Evrogen, Moscow, Russia). The reaction was performed on a QuantStudiotm 5 Real-Time PCR System amplifier (Thermo, Waltham, MA, USA) using the following parameters: 40 cycles of 95 °C-20 s and 60 °C-30 s.

### 4.9. Statistical Analysis

Statistical analysis of hematological parameters, white blood cell differential, and bone marrow differential was performed using parametric one-way analysis of variance with Tukey’s post-test using Prism v8 (GraphPad, Boston, MA, USA). For quantitative data, a test for normality of distribution was performed using the Shapiro–Wilk test. In case of normal distributions, ANOVA with a Tukey post-test was applied. NMR spectra data were analyzed using the Mann–Whitney criterion and ROC AUC calculation in MatLab (The MathWorks, Inc., Natick, MA, USA). Pairwise comparison by Mann–Whitney test was used to assess the difference between groups for the concentration of metabolites in red blood cells. Standard QuantStudioTTM Design and Analysis Software v1.5.2 was used to evaluate gene expression. Further, standard software available online was used for statistical evaluation https://dataanalysis2.qiagen.com/pcr (Qiagen, Hilden, Germany). This software allows online statistical analysis and evaluation of gene expression level using ∆∆Ct approach, applying normalization approaches based on house-kicking genes. Statistical evaluation of significance of changes in gene expression level is based on “*p*-Value”. The “*p*-Value” is calculated based on a Student’s *t*-test of the replicate 2–∆Ct values (or linearized normalized gene expression levels) for each gene in each “control group” and test group comparison. The *p*-value calculation used is based on parametric, unpaired, two-sample equal variance, two-tailed distribution—a method widely accepted in scientific literature. Both groups in each pairwise comparison must contain at least 3 samples for the software to calculate *p*-values for that comparison. Descriptive statistics (mean, standard error) are presented for all quantitative data and shown in figures and summary tables. Statistically significant differences were considered at *p* < 0.05.

## 5. Conclusions

P188 has a hematopoietic stimulating effect in CBA mice and has a dose-dependent effect in restoring the levels of hemoglobin, red blood cell count, reticulocyte count, and hematocrit level. P188 at a dose of 1000 mg/kg brings red blood cell 2,3-BPG concentration close to control levels after carboplatin treatment.

## Figures and Tables

**Figure 1 ijms-26-07081-f001:**
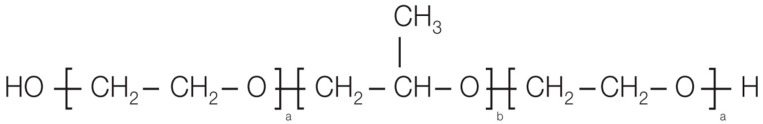
Structural formula of Poloxamer 188. Note: a = 80, b = 27 [9].

**Figure 2 ijms-26-07081-f002:**
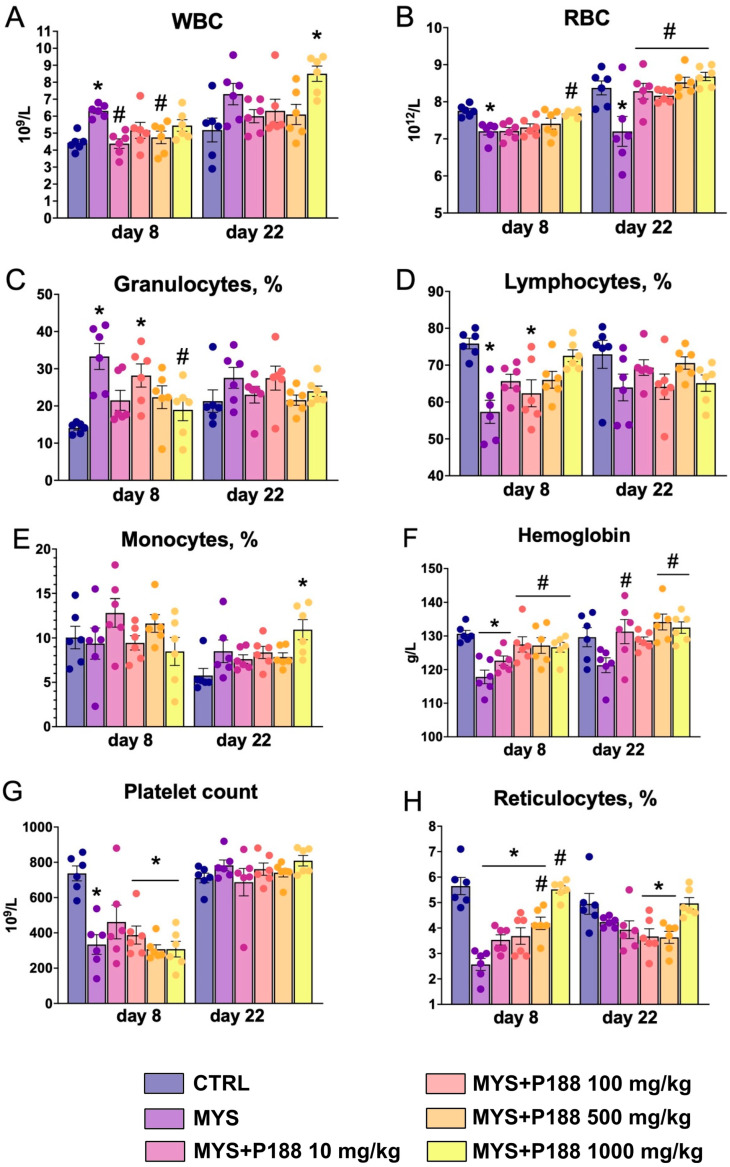
Hematological parameters. (**A**) White blood cell count, (**B**) red blood cell count, (**C**) granulocytes percentage, (**D**)lymphocytes percentage, (**E**) monocytes percentage, (**F**) hemoglobin level, (**G**) platelet count, (**H**) reticulocytes percentage. * *p* < 0.05 compared to the CTRL group; # *p* < 0.05 compared to the MYS group (ANOVA with the Tukey post hoc test).

**Figure 3 ijms-26-07081-f003:**
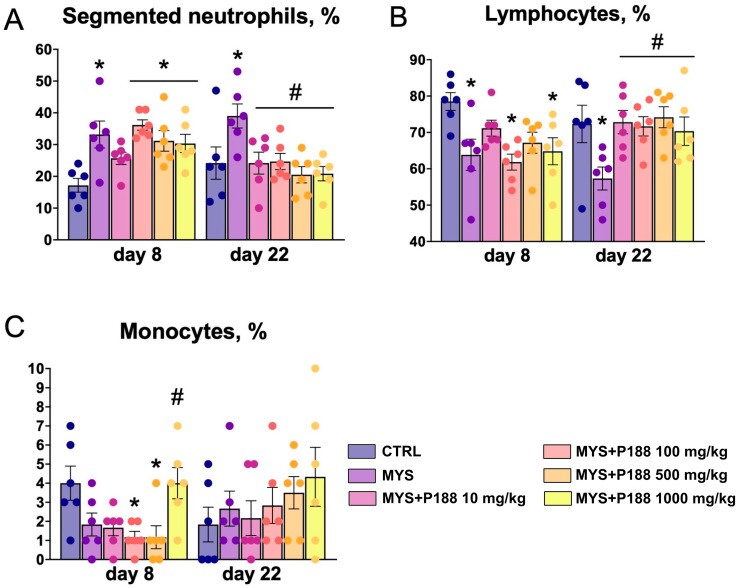
White blood cell differential. (**A**) Segmented neutrophils percentage, (**B**) lymphocytes percentage, (**C**) monocytes percentage. * *p* < 0.05 compared to the CTRL group; # *p* < 0.05 compared to the MYS group (ANOVA with the Tukey post hoc test).

**Figure 4 ijms-26-07081-f004:**
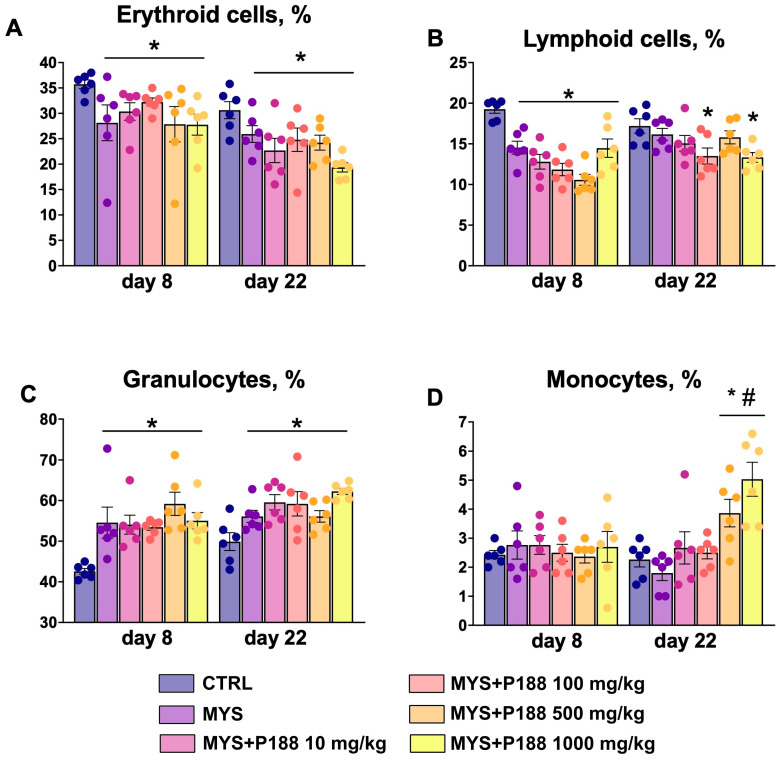
Bone marrow differential. Percentage of (**A**) hematopoietic erythroid cells, (**B**) lymphoid cells, (**C**) granulocytes, and (**D**) monocytes in 500 bone marrow cells. * *p* < 0.05 compared to the CTRL group; # *p* < 0.05 compared to the MYS group (ANOVA with the Tukey post hoc test).

**Figure 5 ijms-26-07081-f005:**
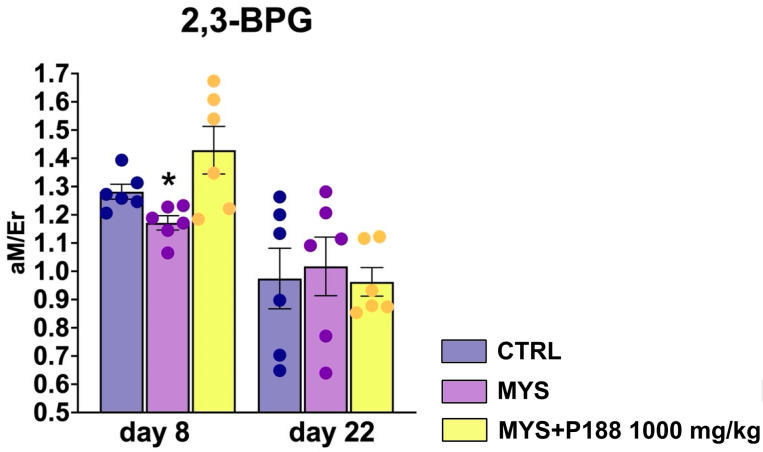
2,3-BPG levels in RBC on the 8th and 22nd day. * *p* < 0.05 compared to the CTRL group (Mann–Whitney test).

**Figure 6 ijms-26-07081-f006:**
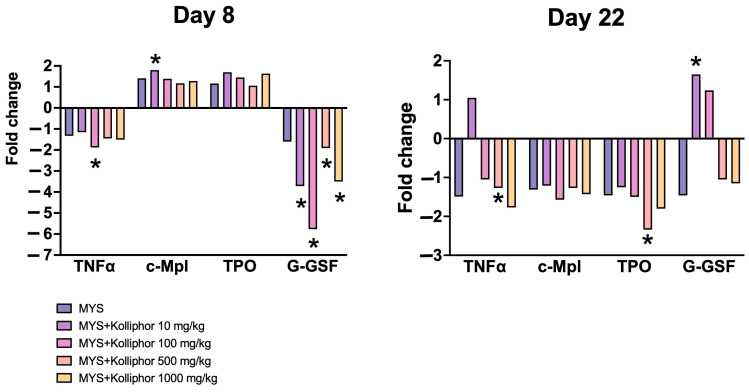
Gene expression fold changes compared to the CTRL group. * *p* < 0.05 compared to the CTRL group (Student’s *t*-test).

**Table 1 ijms-26-07081-t001:** Experimental groups.

Groups	Myelosuppression at Day 0	Treatment from Day 1 to Day 21
1-CTR (Control)	Sterile water	Milli-RO Millipore water 5 mL/kg
2-MYS (Myelosuppression)	Carboplatin at day 0 100 mg/kg, 10 mL/kg i/*p*	Milli-RO Millipore water 5 mL/kg
3-MYS + P188 10 mg/kg		P188 10 mg/kg
4-MYS + P188 100 mg/kg		P188 100 mg/kg
5-MYS + P188 500 mg/kg		P188 500 mg/kg
6-MYS + P188 1000 mg/kg		P188 1000 mg/kg

## Data Availability

The original contributions presented in this study are included in the article. Further inquiries can be directed to the corresponding author.

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
