# Peer review of "Efficacy of Poloxamer 188 in Experimental Myelosuppression Model Induced by Carboplatin in CBA Mice"

_ijms, 2025, doi:10.3390/ijms26157081_

Round 1

Reviewer 1 Report

Comments and Suggestions for Authors

The authors present a well-designed study to investigate the efficacy of Poloxamer 188 to mitigate carboplatin-induced myelosuppression in mice. Overall, the manuscript is well-structured. However, there are some suggestions.

Line 103: Please define “MYS” when it is first mentioned.

Lines 268-275: Considering the species differences, please comment on the potential application of the mouse data for any translational purposes for P188. If the translation is hard to achieve, should other animal models be considered?

Line 285: Are there any reports of potential toxicity of P188? What parameters/biomarkers would be examined for the assessment of the toxicity of repeated dosing of P188?

Only male mice were used in the study. Are there any (potential) sex differences in myelosuppression?

Please elaborate on the dose-dependent effect of P188, especially with the granulocytes percentage and gene expression. The group dosed with 100 mg/kg P188 sometimes had better results than the 1000 mg/kg P188 group.

Author Response

Comments 1: Line 103: Please define “MYS” when it is first mentioned.

Response 1: Thank you for your comment. “MYS” has been defined in the Introduction.

Comments 2: Lines 268-275: Considering the species differences, please comment on the potential application of the mouse data for any translational purposes for P188. If the translation is hard to achieve, should other animal models be considered?

Response 2: Thank you very much for your comment. Discussions are supplemented by speculation on the validity of the mouse model for data transfer. It is reported that mouse models may be useful in the context of hematopoiesis (https://pmc.ncbi.nlm.nih.gov/articles/PMC3633618/, https://www.exphem.org/article/S0301-472X(01)00733-0/fulltext, https://doi.org/10.1016/B978-012369454-6/50059-5).

Comments 3:  Line 285: Are there any reports of potential toxicity of P188? What parameters/biomarkers would be examined for the assessment of the toxicity of repeated dosing of P188?

Response 3: Thank you for your comment. A previous study investigated the acute toxicity of P188 in rats (Arshintseva & Pushkin, 2022; cited in our manuscript). Given its potential for long-term oral administration, we are planning a repeated-dose oral toxicity study for this substance. This study will follow OECD Test Guideline 407 and will comprehensively evaluate the following parameters: hematology, clinical biochemistry, and urinalysis; body weight, food intake, and water consumption; reticulocyte counts in bone marrow smears. Additionally, local irritant effects will be assessed, despite no observable irritation during preliminary 21-day administration.

Comments 4:  Only male mice were used in the study. Are there any (potential) sex differences in myelosuppression?

Response 4: Discussion are supplemented by sex differences limitations in the myelosuppression model.

Comments 5: Please elaborate on the dose-dependent effect of P188, especially with the granulocytes percentage and gene expression. The group dosed with 100 mg/kg P188 sometimes had better results than the 1000 mg/kg P188 group.

Response 5: Although we could not identify a clear dose-dependent pattern in the study, there was pronounced efficacy with respect to hemoglobin, erythrocyte and reticulocyte levels at 500 mg/kg and 1000 mg/kg doses of P188. No significant differences were found in the percentage of granulocytes between groups with P188 administration at different doses. The 100 mg/kg dose was indeed more effective in reducing TNFα on day 22 and G-CSF expression levels on day 8. In terms of clinical parameters, a significant decrease in WBC count and lymphocyte percentage were observed in this group. However, no benefit was seen in hematopoietic parameters in mice treated with P188 at a dose of 100 mg/kg, nor with higher doses of 500 mg/kg or 1000 mg/kg.

Reviewer 2 Report

Comments and Suggestions for Authors

Arina V. Kholina et al. reported an interesting work about the effect of P188 on myelosuppression regulation. The topic fell within the scope of IJMS. The data was informative. The paper should be reconsidered after a Minor Revision. Detailed comments:

  1. Please mention the critical data with p values in the Abstract.
  2. The molecular structure of P188 should be shown.
  3. It must be understood that P188 was a polymeric material, which had a relatively broad distribution of molecular weights. Generally speaking, P188 produced by different manufacturer would have different molecular weight distribution, which might directly impact the pharmacological activities. Please comment on this issue.
  4. Why were the experiments conducted on Day 8 and 22?
  5. P188 at a dose of 1000 mg/kg was actually very high. Was it clinically significant?

Author Response

Comments 1: Please mention the critical data with p values in the Abstract.

Response 1: Thank you for your comment. P values were added to the data presented in the abstract.

Comments 2: The molecular structure of P188 should be shown

Response 2: The molecular structure of P188 has been added to the manuscript.

Comments 3:  It must be understood that P188 was a polymeric material, which had a relatively broad distribution of molecular weights. Generally speaking, P188 produced by different manufacturer would have different molecular weight distribution, which might directly impact the pharmacological activities. Please comment on this issue.

Response 3: Thank you for your comment. We have specified the manufacturer of P188 in the Materials and Methods section and integrated this information into the Discussion.

Comments 4:  Why were the experiments conducted on Day 8 and 22?

Response 4: We chose time points for this study based on a pilot study, where we observed maximal manifestation of myelosuppression (decrease in erythrocytes and hemoglobin primarily) 7 days after carboplatin administration. And 3 weeks after carboplatin administration, recovery of erythrocyte and hemoglobin levels in animals was observed. Thus, we chose the points when the model exhibits the best behavior for our experimental purposes. Also, with this model we had the opportunity to investigate the administration of P188 at 7 and 21 days.

Comments 5: P188 at a dose of 1000 mg/kg was actually very high. Was it clinically significant?

Response 5: We consider the dose of P188 1000 mg/kg to be effective. Notably, the 500 mg/kg dose also achieved significant efficacy, evidenced by rapid (one-week) recovery of reticulocytes and hemoglobin. The 100 mg/kg dose may hold translational relevance. Further investigation using NMR analysis to quantify erythrocyte 2,3-BPG levels at 100 mg/kg and 500 mg/kg is required.